# Sexual Dysfunction in Breast Cancer Survivors: The Role of Clinical, Hormonal, and Psychosocial Factors

**DOI:** 10.3390/healthcare13162061

**Published:** 2025-08-20

**Authors:** Pınar Karaçin, İrem Küçükşahin

**Affiliations:** 1Department of Obstetrics and Gynecology, Dr. Abdurrahman Yurtaslan Ankara Oncology Research and Training Hospital, Ankara 06200, Türkiye; 2Department of Medical Oncology, Dr. Abdurrahman Yurtaslan Ankara Oncology Research and Training Hospital, Ankara 06200, Türkiye; dr_iremoner@hotmail.com

**Keywords:** breast cancer, depression, sexual dysfunction, Female Sexual Function Index, risk factor

## Abstract

**Background and Objectives:** This study aims to investigate female sexual dysfunction (FSD) and the clinical, pathological, and social factors affecting it in women diagnosed with non-metastatic breast cancer. **Materials and Methods:** The study included patients over the age of 18 who were followed up between January 2020 and December 2024, diagnosed with breast cancer at least 12 months previously, and treated. The Female Sexual Function Index (FSFI) and its six subheadings (desire, arousal, lubrication, orgasm, satisfaction, and pain) were used to assess sexual dysfunction, and the Hospital Anxiety and Depression Scale (HADs) was used to assess depression. **Results:** FSD was identified in 86.6% of the 217 patients (mean FSFI score: 20.38). Among women undergoing breast cancer treatment, being over 45 years of age (*p* = 0.003) and the use of luteinizing hormone-releasing hormone (LHRH) agonists (*p* < 0.001) were significantly associated with reduced sexual desire. Conversely, premenopausal status (*p* = 0.012) was associated with increased sexual satisfaction. Independent risk factors for FSD included age, menopausal status, use of LHRH agonists, income level, and axillary dissection. Furthermore, depression was found to have a significant negative impact on sexual desire, lubrication, and orgasm. **Conclusions:** This study demonstrates that sexual dysfunction is common among women undergoing treatment for breast cancer and is influenced by numerous clinical and social factors. These findings highlight the need for strategic interventions to reduce the adverse effects of treatment processes on the sexual health of women with breast cancer.

## 1. Introduction

Breast cancer represents the most commonly diagnosed malignancy worldwide, with an annual incidence exceeding two million new cases, as reported by the World Health Organization [1]. Furthermore, it persists as a primary cause of cancer-related deaths among women on a global scale. In the United States, breast cancer is the most prevalent cancer among women and the second leading cause of cancer-related mortality [2].

Routine immunohistochemical assessment of estrogen receptor (ER), progesterone receptor (PR), and human epidermal growth factor receptor 2 (HER2) expression in breast cancer holds great significance due to the critical prognostic and therapeutic implications of these biomarkers [3]. Clinically, these markers are used to classify breast cancer into distinct subtypes, each characterized by unique features, treatment responses, and disease courses, thereby enabling personalized therapeutic strategies [4]. Approximately 70% of breast cancers are classified as hormone receptor (HR)-positive/HER2-negative, while 15–20% are HER2-positive, and around 15% are triple-negative [5]. In HR-positive breast cancer cases, adjuvant endocrine therapy (ET), typically administered as tamoxifen or aromatase inhibitors, plays a pivotal role following primary treatments such as surgery, radiotherapy, or chemotherapy. In contrast, triple-negative breast cancer generally does not respond to ET [6]. Studies have confirmed that both tamoxifen and aromatase inhibitors effectively reduce recurrence and mortality rates. Integration of ET into systemic therapy for early-stage breast cancer can improve 10-year survival rates up to 92% [7].

Although the primary objective in managing early-stage breast cancer remains to improve overall survival rates, the enhancement of quality of life has increasingly become a crucial aspect of comprehensive, multidisciplinary care. The disease’s progression and the potential adverse effects of treatment can substantially affect patients’ well-being; therefore, treatment strategies should encompass not only physical health but also psychological and social factors.

A breast cancer diagnosis and its subsequent treatment are frequently regarded as traumatic, primarily due to their adverse effects on body image and sexual well-being. Research indicates that the incidence of sexual dysfunction in women undergoing follow-up care post-breast cancer diagnosis can be as high as 73.4% [8,9]. Evaluating female sexual health is, therefore, crucial for the prompt detection and management of sexual dysfunctions. Female sexual dysfunction (FSD) includes a range of issues affecting different phases of the sexual response cycle, such as desire, arousal, orgasm, and pain during intercourse, known as dyspareunia [10,11]. Interventions such as surgery, chemotherapy, radiotherapy, and ET have the potential to disrupt elements vital for preserving sexual health and may impair sexual function by altering hormonal equilibrium [12].

Beyond FSD, a significant proportion of women diagnosed with breast cancer also report experiencing fatigue, depression, and/or anxiety. Established risk factors for anxiety and depression within this population encompass a prior history of mood disorders, younger age at diagnosis, inadequate social support networks, pronounced physical symptoms, active cancer treatment regimens, pharmacologic interventions [13], apprehension regarding mortality and recurrence, altered body image, and disturbed perceptions of femininity, sexuality, and attractiveness [14]. These symptoms can lead to diminished functional capacity and a decline in overall quality of life [15].

Existing research underscores that the sexual health requirements of individuals diagnosed with breast cancer frequently remain unaddressed, and conversations regarding sexuality with healthcare professionals are infrequent [16,17]. This can result in diminished sexual self-efficacy and adversely impact the quality of life for women encountering sexual challenges [18]. While the correlation between depression and FSD is well-documented within the general population, depression—which is reported in up to 32.2% of breast cancer patients—may also be a contributing factor to sexual dysfunction in this specific cohort. Furthermore, studies have indicated that up to 40% of patients experiencing disease recurrence report symptoms of anxiety or depression. Among those with advanced-stage breast cancer, this proportion can reach as high as 42% [14].

Nevertheless, the precise effects of treatment-related elements, including ETs, on sexual function remain understudied. This research endeavors to expand the existing literature by investigating the association between sexual dysfunction and clinical, pathological, and social determinants in women diagnosed with non-metastatic breast cancer. It places particular emphasis on the influence of depression and treatment-related variables on sexual function, thereby filling a void in the current understanding and informing future research endeavors.

The primary hypothesis guiding this study posits that levels of depression and anxiety, treatment-related factors such as ET, and a range of demographic characteristics are correlated with sexual dysfunction. Ultimately, this research aims to deepen our understanding of the etiology of FSD in breast cancer patients and to facilitate the development of more effective treatment approaches that enhance patients’ overall quality of life.

## 2. Materials and Methods

### 2.1. Participants

This study encompassed patients aged 18 years and older who were monitored at our institution between January 2020 and December 2024, diagnosed with early-stage (Stage I–II) or locally advanced (Stage III) HR-positive breast cancer, and had completed chemotherapy, radiotherapy, or targeted therapy. All HR-positive patients had been undergoing ET for a minimum of one year, while patients with triple-negative breast cancer were under observation without systemic treatment.

The Female Sexual Function Index (FSFI) and the Hospital Anxiety and Depression Scale (HADS) were administered via in-person interviews conducted by a clinician. Additional clinical data were retrieved from the hospital’s electronic health record system. The subsequent section details the flow diagram of participant inclusion and exclusion (Figure 1).

In our study, considering the cultural context, the presence of a sexual partner was used as an indicator of sexual activity. This approach is supported by previous studies emphasizing that sexual partner status is an important factor in the assessment of female sexual function [19].

### 2.2. FSFI Calculation

The FSFI is a measurement tool used in the multidimensional evaluation of sexual function in women. Validation for Turkish society was made by Kaya et al. The FSFI scale consists of 19 items in total and is calculated under the six subheadings of desire, arousal, lubrication, orgasm, satisfaction, and pain, and as a total score [20]. Responses given to each item are scored with a Likert-type scale (ranging from 0 or 1 to 5). The scale, which questions experiences in the “last 4 weeks”, is evaluated as 0 points if there is no sexual activity in the relevant period. Sub-dimension scores are calculated by adding the scores of the items belonging to the relevant sub-dimension and then multiplying them by the coefficients determined for the sub-dimension. These coefficients are as follows:Desire: Total score (items 1 and 2) × 0.6;Arousal: Total score (items 3, 4, 5, and 6) × 0.3;Lubrication: Total score (items 7, 8, 9, and 10) × 0.3;Orgasm: Total score (items 11, 12, and 13) × 0.4;Satisfaction: Total score (items 14, 15, and 16) × 0.4;Pain: Total score (items 17, 18, and 19) × 0.4.

The total FSFI score is obtained by summing the subscale scores. The scale total score varies between a minimum of 2 points and a maximum of 36 points. In the literature, scores of 26.55 and below are considered the threshold value indicating FSD [21].

### 2.3. The Hospital Anxiety and Depression Scale (HADS)

The Hospital Anxiety and Depression Scale (HADS) is a validated 14-item self-assessment tool designed to evaluate anxiety (HADS-A) and depression (HADS-D) in outpatient settings, with each item scored from 0 to 3 for a maximum subscale score of 21 [22]. The Turkish version, translated and validated by Aydemir et al., has demonstrated proven reliability and validity in the Turkish population [23]. On the depression subscale, a score of ≤8 was interpreted as suggestive of depression. This threshold is a recognized standard supported by existing research, including studies involving individuals with breast cancer [24].

### 2.4. Statistical Analyses

Statistical analyses were performed using SPSS version 24 and Microsoft^®^ Excel^®^ 2019 MSO 32-bit. Data distribution was assessed for normality using the Kolmogorov–Smirnov test. For comparisons of non-normally distributed continuous variables between two groups, the Mann–Whitney U test was employed. For comparisons across three groups, the Kruskal–Wallis test was used. Given the non-normal distribution of variables, Spearman’s rank correlation was used to examine associations between HADS-D and FSFI subscale scores. The Kolmogorov–Smirnov test was used to confirm deviations from normality.

Potential confounding factors, including age, education level, and disease duration, were considered in the analyses.

Correlation coefficients were interpreted according to the following criteria:Perfect correlation: ±1;Strong correlation: ±0.50 to ±1.00;Moderate correlation: ±0.30 to ±0.49;Weak correlation: ±0.00 to ±0.29;No correlation: 0.

To determine independent predictors of FSD (defined as an FSFI score ≤ 26.55), binary logistic regression analysis was conducted. Multivariate modeling was performed using the Forward: LR method. Statistical significance was defined as a *p*-value < 0.05 and a Type I error rate < 5%.

## 3. Results

Women aged ≤45 years exhibited significantly higher sexual desire scores (3.4 ± 1.2; *p* = 0.003). Additionally, pre- or perimenopausal participants had higher satisfaction scores (*p* = 0.012), nulliparous women showed higher orgasm scores (*p* = 0.008), and women not receiving LHRH agonists or adjuvant endocrine therapy reported higher sexual desire scores (*p* < 0.001 and *p* = 0.014, respectively) (Table 1).

In total, 60.8% (*n* = 132) of participants had completed high school. A statistically significant difference in sexual desire scores was observed with educational level (*p* = 0.009) (Table 2).

A weak inverse correlation was found between HADS-D scores and sexual desire (r = −0.187; *p* = 0.006), lubrication (r = −0.146; *p* = 0.032), and orgasm (r = −0.157; *p* = 0.020). Moderate positive correlations were observed between sexual desire and arousal (r = 0.291; *p* < 0.001), lubrication (r = 0.340; *p* < 0.001), and pain (r = 0.455; *p* < 0.001) (Table 3).

The FSFI, 86.6% of women (*n* = 188) were identified with sexual dysfunction (score < 26.55), with a mean score of 20.38 ± 5.57. Univariate logistic regression showed that sexual dysfunction risk was significantly higher in women aged ≤45 years (OR: 2.47; *p* = 0.035), those receiving LHRH agonists (OR: 4.00; *p* = 0.004), with higher monthly income (OR: 2.56; *p* = 0.007), undergoing axillary dissection (OR: 2.60; *p* = 0.004), and receiving adjuvant endocrine therapy (OR: 3.78; *p* = 0.033). Religion was not assessed (Table 4).

In multivariate analysis, age, monthly income, menopausal status, axillary dissection, and use of LHRH agonists constituted a predictive model for sexual dysfunction (Figure 2).

## 4. Discussion

A key finding of this study is the elevated prevalence of sexual dysfunction among sexually active women with breast cancer undergoing adjuvant ET. Assessment of patients with early-stage or locally advanced breast cancer following completion of systemic treatments revealed that age, menopausal status, use of LHRH agonists, income level, axillary dissection, adjuvant ET, and depression significantly impacted sexual dysfunction. Furthermore, significant associations were observed between depression levels and sexual desire, vaginal lubrication, and orgasm.

Sexual dysfunction is common in the general population, with reported prevalence rates ranging from 5.5% to 77% [25]. It is also frequently observed among women diagnosed with breast cancer; however, varying prevalence rates have been reported across different patient populations due to numerous influencing factors. A meta-analysis reported a prevalence of female sexual dysfunction (FSD) of 73.4% among breast cancer patients, with a mean FSFI score of approximately 19.00 [9]. In a study conducted in Turkey, sexual dysfunction was detected in 55% of breast cancer patients [26]. Another study reported an even higher prevalence of 83.08% among young breast cancer patients [27]. Furthermore, more than 80% of women were reported to experience sexual problems within the first six months of adjuvant ET [28]. In another study focusing on breast cancer patients undergoing adjuvant ET, clinically significant sexual dysfunction was found in 59% of sexually active participants, with 58% of these cases directly attributed to cancer treatment [29]. In our study, sexual dysfunction was identified in 86.6% of women receiving endocrine therapy, and the mean FSFI score was 20.38. A recent extensive cohort study also demonstrated that the prevalence of sexual dysfunction was significantly higher in women diagnosed with breast cancer compared to those without a history of cancer. According to data from the CONSTANCES cohort, 42.5% of breast cancer patients reported sexual dysfunction; this association remained statistically significant even after adjusting for age, educational level, and other sociodemographic factors [30]. These findings suggest that sexual dysfunction may result not only from the treatments administered but also from the psychosocial burden associated with the diagnosis itself. The higher prevalence observed in our study may be attributable to the fact that our sample was limited to women who were both receiving adjuvant ET and sexually active.

Endocrine therapies such as aromatase inhibitors, tamoxifen, and LHRH agonists, which exert their effects primarily through estrogen suppression, are associated with a range of sexual side effects. These adverse effects commonly include vaginal dryness, dyspareunia, decreased libido, and difficulty achieving orgasm [31]. A study conducted in Sweden among postmenopausal breast cancer patients demonstrated an increased frequency of sexual dysfunction symptoms, particularly vaginal dryness and dyspareunia, in those receiving ET [32]. Similarly, another study reported significantly higher rates of reduced libido and vaginal dysfunction in patients undergoing endocrine treatment, with notably high prevalence rates of inadequate vaginal lubrication (73.9%) and dyspareunia (56.5%) [33], clearly indicating the suppressive impact of endocrine therapy on sexual function. Consistent with these findings, our study revealed that patients receiving LHRH agonists had significantly lower scores in the sexual desire and satisfaction domains compared to those not receiving these agents. Furthermore, a negative correlation was observed between depression levels and both vaginal lubrication and orgasm scores. Interestingly, although the frequency of dyspareunia was higher in the endocrine therapy group, the difference did not reach statistical significance. This may be attributed to several factors, including sample size limitations, individual variations in pain thresholds, or the sensitivity of the measurement tools used.

In our study, both the type of surgical treatment and age were found to be significantly associated with sexual dysfunction. Surgical approaches, particularly those involving breast-conserving surgery or reconstruction, may influence body image and, consequently, sexual functioning. While some studies in the literature suggest that the type of surgery does not directly affect sexual health [34], our findings indicate that more invasive procedures, such as axillary dissection, may negatively impact sexual life. This may be attributed to both physical and psychosocial consequences, including restricted arm movement, increased risk of lymphedema, and disturbances in body image. Similarly, age was found to be significantly correlated with sexual dysfunction in both univariate and multivariate analyses. Although one study reported that age alone was not a significant predictor of the prevalence of sexual dysfunction, it did indicate a substantial decline in sexual desire with increasing age [29]. Our findings underscore the importance of considering age as a relevant factor in the assessment of sexual function during the survivorship phase of breast cancer.

The diagnosis and treatment process of breast cancer involves invasive interventions that directly affect the female body and are frequently accompanied by psychiatric issues, rendering it a complex condition. Depression is one of the most commonly encountered psychiatric disorders among women diagnosed with breast cancer, with a reported prevalence reaching up to 30% [35]. Notably, a strong negative correlation has been observed between depression levels and specific domains of sexual function, such as sexual desire, vaginal lubrication, and orgasm, clearly indicating the significant impact of psychological status on sexual health [36]. Consistent with these findings, our study also revealed a strong association between depression levels and sexual function. Specifically, negative correlations were identified between depression scores and the subdomains of the Female Sexual Function Index, particularly sexual desire, lubrication, and orgasm. According to the Hospital Anxiety and Depression Scale—depression subscale (HADS-D), women exhibiting depressive symptoms had a 3.3-fold higher risk of experiencing sexual dysfunction. These findings are consistent with previous research demonstrating a significant relationship between depressive symptoms and sexual dysfunction [35]. Moreover, several studies have reported a significant association between partner status and women’s sexual self-efficacy, ability to cope with sexual problems, and psychosocial quality of life. In particular, women without a sexual partner reported lower levels of self-efficacy, more pronounced sexual difficulties, and poorer psychosocial quality of life [28]. Although partner status was not directly evaluated in our study, variations in sexual function were observed according to factors such as menopausal status and age.

This study has several limitations. First, its single-center and cross-sectional design may limit the generalizability of the findings to the broader breast cancer population. The inclusion criterion of being sexually active with a sexual partner in the past three months may have led to an underestimation of the true prevalence of sexual dysfunction in the general population. Additionally, the exclusion of individuals with psychiatric disorders or insufficient proficiency in Turkish may have resulted in a sample comprising individuals with better mental health and communication skills, potentially limiting the evaluation of psychological determinants. The lack of detailed subgroup analysis according to different types of endocrine therapy is another limitation that may have hindered the clear identification of specific effects. Furthermore, although adjuvant systemic treatments were generally administered according to clinical guidelines, variations in treatment intensity according to disease stage were not thoroughly evaluated, which should also be acknowledged.

Despite these limitations, the study also possesses several strengths. Data were collected through face-to-face interviews, and validated assessment tools were employed, both of which contribute significantly to the reliability of the study.

## 5. Conclusions

Our study demonstrated associations between age, menopausal status, LHRH use, income level, axillary dissection, adjuvant ET, depression, and sexual dysfunction. For modifiable risk factors such as LHRH use and adjuvant ET, close monitoring and, when necessary, adjustment of the treatment plan may be warranted. Future research should focus on identifying interventions that can reduce sexual dysfunction in these patients.

## Figures and Tables

**Figure 1 healthcare-13-02061-f001:**
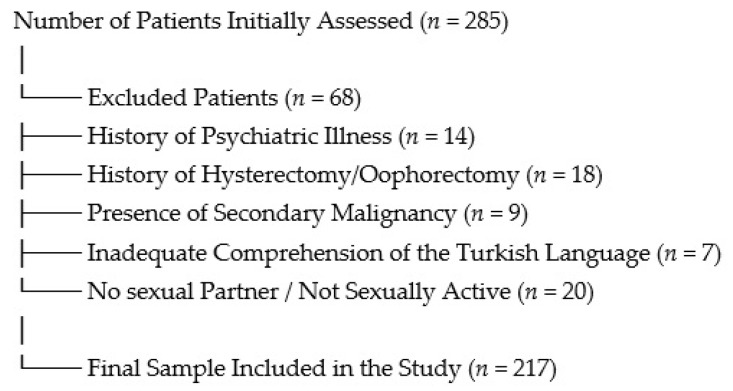
Flow diagram of study participant selection.

**Figure 2 healthcare-13-02061-f002:**
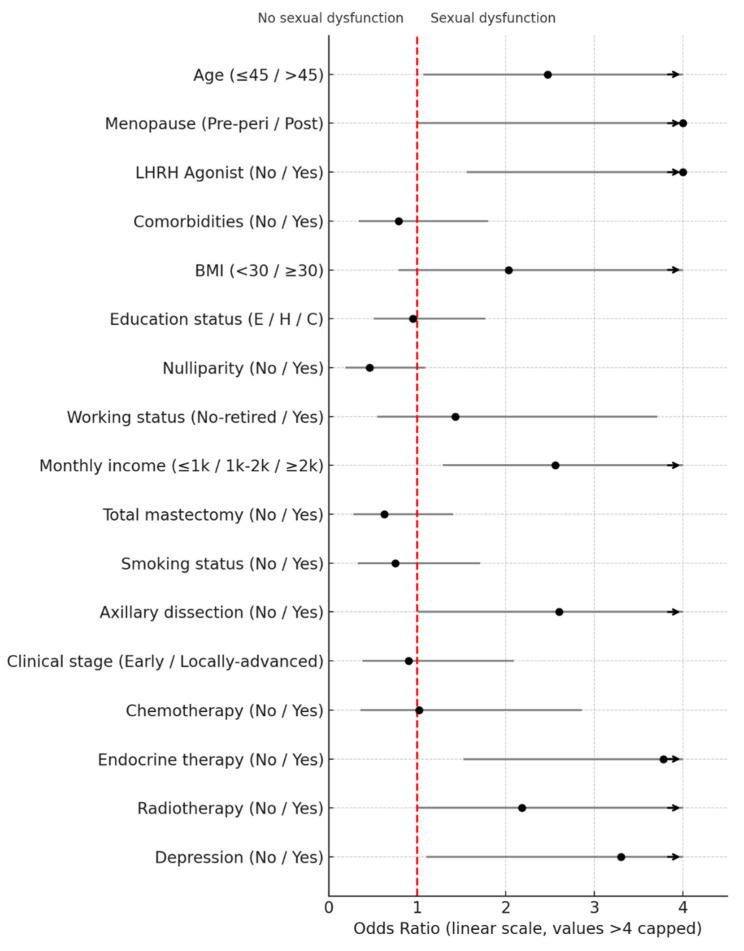
Evaluation of factors predicting the presence of sexual dysfunction using a multivariate logistic regression model.

**Table 1 healthcare-13-02061-t001:** Mean ± standard deviation (SD) values of the six subscales of the Female Sexual Function Index (FSFI) and the *p*-values of statistical comparisons according to clinical characteristics.

	*n*	%	Desire	Arousal	Lubrication	Orgasm	Satisfaction	Pain
Age	*p*-values *		0.003	0.260	0.033	0.407	0.152	0.076
≤45	109	50.2	3.4 ± 1.2	2.4 ± 1.5	3.3 ± 1.1	3.2 ± 1.3	2.7 ± 1.7	3.2 ± 1.9
>45	108	49.8	2.9 ± 1.1	2.3 ± 1.6	3.0 ± 1.3	3.0 ± 1.3	2.3 ± 1.8	2.6 ± 2.1
Comorbidities	*p*-values *		0.077	0.191	0.387	0.763	0.815	0.131
No	65	30.0	3.2 ± 1.3	2.5 ± 1.4	3.1 ± 1.2	3.0 ± 1.8	2.8 ± 1.7	2.8 ± 2.0
Yes	152	70.0	2.8 ± 1.3	2.0 ± 1.6	2.9 ± 1.2	2.6 ± 2.0	2.4 ± 1.8	2.5 ± 2.0
Menopause	*p*-values *		0.103	0.857	0.372	0.872	0.012	0.706
Pre-peri	176	81.1	3.2 ± 1.2	2.3 ± 1.6	3.1 ± 1.2	2.9 ± 1.8	2.8 ± 1.8	2.6 ± 1.8
Post	41	18.9	2.8 ± 1.5	2.3 ± 1.6	3.0 ± 1.3	2.8 ± 1.9	2.6 ± 1.8	2.6 ± 2.0
BMI	*p*-values *		0.059	0.811	0.914	0.236	0.998	0.143
<30	146	67.3	3.4 ± 1.2	2.9 ± 1.2	3.2 ± 1.2	3.2 ± 1.7	2.9 ± 1.8	2.4 ± 1.8
≥30	71	32.7	2.9 ± 1.4	2.4 ± 1.6	3.1 ± 1.2	2.4 ± 1.9	2.4 ± 1.8	2.9 ± 2.0
Nulliparity	*p*-values		0.261	0.595	0.205	0.008	0.753	0.243
No	176	81.1	3.1 ± 1.3	2.4 ± 1.5	3.1 ± 1.2	2.8 ± 1.8	2.6 ± 1.7	2.6 ± 2.0
Yes	41	18.9	3.1 ± 1.2	2.3 ± 1.6	3.0 ± 1.2	2.9 ± 1.9	2.7 ± 1.8	2.7 ± 2.0
Total mastectomy	*p*-values **		0.089	0.810	0.572	0.869	0.502	0.516
No	147	67.7	3.1 ± 1.3	2.4 ± 1.5	3.1 ± 1.2	2.9 ± 1.8	2.7 ± 1.8	2.7 ± 2.0
Yes	70	32.3	3.1 ± 1.2	2.3 ± 1.6	3.0 ± 1.2	2.8 ± 1.9	2.6 ± 1.7	2.6 ± 2.0
Axillary dissection	*p*-values *		0.120	0.636	0.738	0.865	0.416	0.601
No	135	62.2	3.2 ± 1.3	2.4 ± 1.5	3.1 ± 1.2	2.9 ± 1.8	2.7 ± 1.7	2.7 ± 2.0
Yes	82	37.8	3.0 ± 1.2	2.4 ± 1.6	3.0 ± 1.2	2.8 ± 1.9	2.6 ± 1.8	2.6 ± 2.0
Clinical stage	*p*-values *		0.054	0.396	0.869	0.935	0.594	0.890
Early	154	71.0	3.3 ± 1.2	2.6 ± 1.4	3.2 ± 1.1	3.0 ± 1.7	2.8 ± 1.7	2.8 ± 1.9
Locally advanced	63	29.0	2.9 ± 1.4	2.3 ± 1.6	3.0 ± 1.3	2.7 ± 2.0	2.6 ± 1.8	2.6 ± 2.1
Chemotherapy	*p*-values *		0.954	0.260	0.615	0.374	0.619	0.422
No	37	17.1	3.1 ± 1.4	2.3 ± 1.5	3.0 ± 1.2	3.1 ± 1.9	2.8 ± 1.8	2.5 ± 2.0
Yes	180	82.9	3.1 ± 1.2	2.6 ± 1.6	3.1 ± 1.3	2.8 ± 1.8	2.3 ± 1.7	2.7 ± 2.0
LHRH agonist	*p*-values *		<0.001	0.256	0.119	0.280	0.046	0.151
No	115	53.0	3.2 ± 1.2	2.5 ± 1.5	3.2 ± 1.2	3.0 ± 1.7	2.8 ± 1.7	2.8 ± 1.9
Yes	102	47.0	2.8 ± 1.3	2.1 ± 1.6	2.9 ± 1.2	2.6 ± 2.0	2.4 ± 1.8	2.5 ± 2.0
Endocrine therapy	*p*-values *		0.014	0.607	0.150	0.970	0.097	0.346
No	29	13.4	3.3 ± 1.2	2.6 ± 1.4	3.2 ± 1.2	3.1 ± 1.7	2.9 ± 1.7	2.9 ± 1.9
yes	188	86.6	2.9 ± 1.3	2.2 ± 1.6	3.0 ± 1.2	2.6 ± 2.0	2.4 ± 1.8	2.5 ± 2.0
Radiotherapy	*p*-values *		0.429	0.150	0.603	0.744	0.408	0.429
No	77	35.5	3.0 ± 1.1	2.2 ± 1.5	3.0 ± 1.3	2.7 ± 1.9	2.3 ± 1.8	2.7 ± 2.1
yes	140	64.5	3.2 ± 1.2	2.5 ± 1.6	3.1 ± 1.2	2.9 ± 1.8	2.6 ± 1.8	2.9 ± 2.0

LHRH: Luteinizing hormone-releasing hormone, * Mann–Whitney U test, ** Kruskal–Wallis test.

**Table 2 healthcare-13-02061-t002:** Mean ± standard deviation (SD) values of the six subscales of the Female Sexual Function Index (FSFI) and the *p*-values of statistical comparisons according to sociodemographic characteristics.

	*n*	%	Desire	Arousal	Lubrication	Orgasm	Satisfaction	Pain
Education	*p*-values **		0.009	0.460	0.693	0.049	0.410	0.387
Elementary school	37	17.1	2.9 ± 1.2	2.2 ± 1.7	3.2 ± 1.0	2.2 ± 2.0	2.4 ± 2.0	2.4 ± 2.0
High school	132	60.8	3.3 ± 1.2	2.4 ± 1.5	3.0 ± 1.3	3.1 ± 1.7	2.4 ± 1.7	2.9 ± 2.0
College or higher	48	22.1	2.8 ± 1.1	2.4 ± 1.6	3.2 ± 1.2	2.8 ± 1.9	2.8 ± 1.9	2.9 ± 2.1
Monthly income (USD)	*p*-values **		0.938	0.323	0.938	0.155	0.712	0.427
≤1000	35	16.1	3.2 ± 1.2	2.8 ± 1.7	3.1 ± 1.1	3.4 ± 1.5	2.4 ± 1.8	2.5 ± 2.0
1000–2000	137	63.1	3.1 ± 1.1	2.3 ± 1.4	3.1 ± 1.2	2.7 ± 1.9	2.6 ± 1.8	3.0 ± 2.1
≥2000	45	20.7	3.1 ± 1.4	2.4 ± 1.8	3.0 ± 1.3	3.1 ± 1.9	2.3 ± 1.7	2.7 ± 2.0
Working status	*p*-values		0.051	0.205	0.029	0.695	0.821	0.055
No-retired	160	73.7	3.2 ± 1.2	2.4 ± 1.5	3.2 ± 1.2	2.8 ± 1.8	2.5 ± 1.8	3.0 ± 1.9
yes	57	26.3	2.9 ± 1.1	2.2 ± 1.4	2.8 ± 1.3	2.8 ± 2.0	2.9 ± 1.7	2.4 ± 2.0
Smoking status	*p*-values *		0.562	0.932	0.622	0.335	0.779	0.530
No	63	29.0	3.3 ± 1.2	2.6 ± 1.5	3.2 ± 1.2	3.1 ± 1.7	2.9 ± 1.7	2.9 ± 1.9
Yes	154	71.0	2.8 ± 1.3	2.0 ± 1.6	2.9 ± 1.2	2.6 ± 2.0	2.4 ± 1.8	2.5 ± 2.0

* Mann–Whitney U test, ** Kruskal–Wallis test.

**Table 3 healthcare-13-02061-t003:** Correlation analysis between the HADS–depression subscale and the subdomains of the Female Sexual Function Index (FSFI).

		Desire	Arousal	Lubrication	Orgasm	Satisfaction
HADS-D	−0.187	−0.070	−0.146	−0.157	−0.061	−0.091
*p*-value	0.006	0.305	0.032	0.020	0.372	0.180
Desire		0.291	0.340	0.036	0.056	0.455
*p*-value		<0.001	<0.001	0.599	0.413	<0.001
Arousal			−0.192	0.018	−0.406	−0.131
*p*-value			0.004	0.788	<0.001	0.053
Lubrication				−0.051	0.428	0.604
*p*-value				0.455	<0.001	<0.001
Orgasm					−0.018	−0.009
*p*-value					0.793	0.890
Satisfaction						0.416
*p*-value						<0.001

HADS-D: Hospital Anxiety and Depression Scale—depression subscale, FSFI: Female Sexual Function Index (desire, arousal, lubrication, orgasm, satisfaction, pain).

**Table 4 healthcare-13-02061-t004:** Evaluation of factors predicting the presence of sexual dysfunction using univariate logistic regression analysis.

Variable	Categories	OR (95% CI)	*p*-Value
Age	≤45/>45	2.47 (1.07–5.71)	0.035
Menopause	Pre–peri/Post	7.57 (1.00–57.33)	0.050
LHRH agonist	No/Yes	4.00 (1.56–10.27)	0.004
Monthly income (USD)	≤1 k/1 k–2 k/≥2 k	2.56 (1.29–5.05)	0.007
Axillary dissection	No/Yes	2.60 (1.01–6.69)	0.047
Endocrine therapy	No/Yes	3.78 (1.52–9.42)	0.004
Depression	No/yes	3.30 (1.10–9.90)	0.033

LHRH: Luteinizing hormone-releasing hormone, ≤1 k/1 k–2 k/≥2 k: ≤1000/1000–2000/≥2000.

## Data Availability

The data presented in this study are available on request from the corresponding author.

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
