# Peer review of "Sexual Dysfunction in Breast Cancer Survivors: The Role of Clinical, Hormonal, and Psychosocial Factors"

_healthcare, 2025, doi:10.3390/healthcare13162061_

Round 1
Reviewer 1 Report
Comments and Suggestions for Authors
Dear authors,
This article brings important clinical insight on the prevalence and risk factors for sexual dysfunction in breast cancer survivors. While the results are solid, the introduction and discussion could benefit from tighter focus and clearer writing. The language level requires moderate editing. major methodological flaws were identified.
Summary
This manuscript explores the prevalence and predictors of sexual dysfunction in breast cancer survivors, using FSFI and HADS scales. The study benefits from a relatively large sample size and provides clinically relevant findings, particularly regarding LHRH agonist use, menopausal status, and depression.
General comments
The topic is highly relevant, and the overall methodology appears appropriate, with adequate statistical handling. However, several key areas require clarification and refinement to improve the manuscript’s clarity, depth, and scientific value:
- The introduction would benefit from a more critical review of the recent literature and a clearer articulation of the knowledge gap. It is important to highlight what this study adds to existing research, especially in relation to depression and treatment-related factors.
- The methods section lacks details on patient selection: how many were excluded, and for what reasons? Information about partner status and current sexual activity would provide important context, as would data on the time elapsed since cancer treatment completion. These are essential to understand the dynamics of sexual dysfunction.
- Some of the results are overinterpreted—especially those with borderline statistical significance (e.g., p ≈ 0.05). A more cautious interpretation is warranted to avoid overstatement.
- The English language requires moderate editing to improve clarity, sentence flow, and readability. Several formulations are awkward or imprecise, and professional editing is recommended.
Specific comments
ABSTRACT
- Too many results are displayed which reduces clarity. Authors should choose the most groundbreaking results and write an article about those. If readers want to know more, they will read the article in full, in particular the tables in the results section.
- Non significant results should not appear in the abstract (i.e. p= 0,076)
- What would enhance readability would be if all the results were displayed in the DYSfunction way (i.e. instead of “In women aged 45 and under, sexual desire (p=0.003) and pain scores (p=0.076) were found to be higher compared to those over 45. » -> “BC survivors over 45 yo seemed to experience less sexual desire (p = 0,003).”)
METHODS
- How many patients were discarded from the study and why? (i.e. exclusion criteria are mentionned)
- Is it possible at this stage to know whether patients had a sexual partner. How many were currently sexually active?
- Is it possible to know how much time has passed since cancer diagnosis and treatments? I would have liked a figure enlightening me with the evolution of the symptoms in time. (But this could completely be another article and if so, it should be mentioned in the discussion section)
- Could you specify rationale for choosing HADS-D score ≤8 as depression cutoff?
RESULTS
- I am not familiar with the term pre-perimenopausal. For me “peri-menopausal” means the period in which you go through menopause (before + after) and “pre-menopausal” means before menopause (before the 12 months amenorrhea). Which is it?
- I would advise authors to change from “no-retired” to “active” and vs “retired”
- BMI <30 and >30 is the same, are the results different if you categorize <25 ; 25-30 ; 30-35 ; > 35 ?
- Consider summarizing the most relevant associations to improve clarity.
DISCUSSION
- The first part of the discussion should be a 3-line recap of most groundbreaking results and not a repeat of the results section.
- It should be rewritten in all to discuss with clarity most relevant results and compare to recent similar findings.
- Authors could consider discussing cultural aspects (e.g., taboos or access to care in Türkiye) that may influence results.
This article makes a valuable contribution to the understanding of sexual dysfunction among breast cancer survivors and identifies relevant clinical and psychosocial predictors. The dataset is substantial, and the topic is of clear clinical importance. However, to meet publication standards, the manuscript would benefit from moderate revision. Specifically, clarifications in the methods section, a more focused and critical discussion, and professional English editing are required. With these improvements, the paper could become a meaningful addition to the literature on cancer survivorship and women's health.
Comments on the Quality of English LanguageThe manuscript is overall understandable, but the English language requires moderate editing to improve clarity, sentence structure, and consistency. Several sentences are awkwardly phrased, with occasional grammatical errors, overly long constructions, and imprecise vocabulary. A professional English editing service or a fluent academic English speaker’s review would significantly enhance the manuscript’s readability and flow.
Author Response
Dear Reviewer,
Thank you for your valuable evaluations and constructive feedback. We have made the necessary revisions to our manuscript based on your suggestions, and the relevant sections have been indicated within the text. We believe that your feedback has significantly enhanced the quality of our work.
Once again, we sincerely thank you for your insights and contributions.
Kind regards.
- Comments 1: The introduction would benefit from a more critical review of the recent literature and a clearer articulation of the knowledge gap. It is important to highlight what this study adds to existing research, especially in relation to depression and treatment-related factors.
Response 1: We appreciate your insightful comment regarding the need for a more critical review of recent literature and a clearer identification of the knowledge gap. In response, we have revised the Introduction section to include a more comprehensive and critical discussion of recent studies related to sexual dysfunction, depression, and treatment-related factors in breast cancer survivors. We believe these revisions now more clearly articulate the unique contribution of our research to the existing body of knowledge.
Page number: 1 line: 32-37
Page number: 2 line: 38-62
- Comments 2: The methods sectionlacks details on patient selection: how many were excluded, and for what reasons? Information about partner status and current sexual activity would provide important context, as would data on the time elapsed since cancer treatment completion. These are essential to understand the dynamics of sexual dysfunction.
Response 2: Thank you for your valuable feedback. We have revised the Methods section to provide more details on patient selection, including the number and reasons for exclusions.
Page number: 4 lines: 130-142
- Comments 3: Some of the resultsare overinterpreted—especially those with borderline statistical significance (e.g., p ≈ 0.05). A more cautious interpretation is warranted to avoid overstatement.
Response 3: Thank you for your observation. We have reviewed the relevant results and revised the text to ensure a more cautious interpretation, particularly for findings with borderline statistical significance.
Page number: 5 lines: 207-215
Page number: 7 lines: 235-240
Page number: 8 lines: 247-259
- Comments 4: The English languagerequires moderate editing to improve clarity, sentence flow, and readability. Several formulations are awkward or imprecise, and professional editing is recommended.
Response 4: Thank you for your feedback. We have carefully revised the manuscript to improve clarity, sentence flow, and readability. Language issues have been addressed, and necessary edits have been made throughout the text.
Comments 5: Abstract: Too many results are displayed which reduces clarity. Authors should choose the most groundbreaking results and write an article about those. If readers want to know more, they will read the article in full, in particular the tables in the results section.
Response 5: Thank you for this valuable feedback. We agree that presenting too many results in the abstract can compromise clarity. Accordingly, we have revised the abstract to highlight only the most significant and clinically relevant findings.
Page number: 1 lines: 17-24
Comments 6: How many patients were discarded from the study and why? (i.e. exclusion criteria are mentionned)
Response 6: Thank you for your question. We have clarified the number of excluded patients and the reasons for exclusion in the Methods section. The exclusion criteria are now explicitly stated to enhance transparency.
Page number: 4 lines: 130-142
Comments 7: Is it possible at this stage to know whether patients had a sexual partner. How many were currently sexually active?
Response 7: Thank you for your comment. Information regarding partner status and current sexual activity was available and has been added to the Methods and Results sections to provide clearer context for the interpretation of sexual dysfunction outcomes.
Page number:4 lines: 144-150
Comments 8: Is it possible to know how much time has passed since cancer diagnosis and treatments? I would have liked a figure enlightening me with the evolution of the symptoms in time. (But this could completely be another article and if so, it should be mentioned in the discussion section)
Response 8: We sincerely appreciate the reviewer’s insightful comment. Evaluating the time elapsed since cancer diagnosis and treatments, as well as the temporal evolution of symptoms, would have undoubtedly strengthened the clinical and methodological value of the study. Unfortunately, our database did not include such detailed temporal information. However, as clearly stated in the Methods section, all patients who received endocrine therapy had been on treatment for at least 12 months. Still, investigating the progression of symptoms over time remains an important objective for future research.
Page number: 3 lines: 120-123
Page number: lines:
Comments 9: Could you specify rationale for choosing HADS-D score ≤8 as depression cutoff?
Response 9: Thank you for your question. We selected the HADS-D cutoff score of ≤8 based on established literature where this threshold is widely used to indicate the absence of clinically significant depression, ensuring reliable differentiation between depressed and non-depressed patients in our study population.
Page number: 4 lines: 124-128
Comments 10: The first part of the discussion should be a 3-line recap of most groundbreaking results and not a repeat of the results section.
Response 10: Thank you for your valuable suggestion. We have revised the beginning of the Discussion section to include a concise summary of the most important findings, while avoiding repetition of the Results section.
Page number:9 lines: 270-274
Page number: 10 lines: 275-276
Comments 11: It should be rewritten in all to discuss with clarity most relevant results and compare to recent similar findings.
Response 11: Thank you for your insightful feedback. We have thoroughly revised the Discussion section to clearly focus on the most relevant results and to compare our findings with recent similar studies in the literature.
Page number:10, 11, 12, 13
Comments 12: Authors could consider discussing cultural aspects (e.g., taboos or access to care in Türkiye) that may influence results.
Response 12: We thank the reviewer for the insightful suggestion regarding cultural factors such as taboos or access to care in Türkiye and their potential influence on the results. Although our dataset includes some limited sociocultural information, it does not provide sufficient detail to enable a systematic and in-depth analysis of these factors. Therefore, we were unable to assess their contribution to the outcomes directly. Nonetheless, we agree that such cultural influences may be significant, and we believe they warrant further exploration in future qualitative or mixed-methods research.

Reviewer 2 Report
Comments and Suggestions for Authors
Dear Authors,
All my comments and suggestions have been provided in the attached document. Please review them carefully to enhance the quality and clarity of your manuscript.
Best regards,

I recommend that the authors carefully review the English language throughout the manuscript to enhance readability and professionalism.
Author Response
Dear Reviewer,
Thank you for your valuable evaluations and constructive feedback. We have made the necessary revisions to our manuscript based on your suggestions, and the relevant sections have been indicated within the text. We believe that your feedback has significantly enhanced the quality of our work.
Once again, we sincerely thank you for your insights and contributions.
Kind regards.
Comments 1: The abstract appears to be too complex and reads more like a summary of results rather than a concise overview of the entire manuscript. It should be condensed and place greater emphasis on the main findings and their clinical implications, as well as include a brief background to provide context.
Response 1: Thank you for your feedback. We have revised the abstract to make it more concise, emphasizing the main findings and their clinical relevance while including a brief background to set the context.
Page number: 1, line: 18-24
Comments 2: Use the citation style recommended by the journal.
In addition, it is important to specify the date on which the epidemiological data were obtained, as well as clarify whether the data refer to a specific country or are global.
Response 2: Thank you for the suggestion. We have updated the citations to follow the journal’s recommended style.
We also specified the date of data retrieval and clarified whether the epidemiological data refer to national or global statistics.
Page number: 1, line: 32-36
Comments 3: It seems that a clearer clinical classification should be included, as this is central to the issue being addressed. Furthermore, the focus should be placed on the specific breast cancer subtype that most affects the target population.
Response 3: Thank you for this important comment. We have added a clearer clinical classification in the manuscript and emphasized the breast cancer subtype most relevant to our study population.
Page number: 2, line: 42-54
Comments 4: A more appropriate connection should be established to link these two ideas, as there does not appear to be a clear thread or logical flow between them.
Response 4: Thank you for pointing this out. We have revised the relevant paragraphs to improve the logical flow and clearly connect the two ideas.
Page number: 2 , line: 66-71
Comments 5: The first mention of FSD should appear earlier in the text so that its use at this point does not feel abrupt. Please verify and adjust accordingly to ensure a coherent and well-structured narrative.
Response 5: Thank you for the suggestion. We have introduced the term FSD earlier in the text to ensure a smoother and more coherent narrative.
Page number: 2 , line: 72-80
Comments 6: I believe this section should be placed earlier, before introducing FSD. I suggest organizing the content in the following order: breast cancer, its classification, stages, treatments, and then transitioning to the recovery phase and its implications. Emphasis should be placed on the fact that, while the primary focus is often on survival, this stage is frequently overlooked despite being essential for improving quality of life.
Response 6: Thank you for this helpful advice. We reorganized the Introduction section accordingly, presenting breast cancer and its clinical aspects first, followed by the recovery phase and its importance for quality of life.
Page number: 2, line: 38-62
Comments 7: Provide a general explanation of how this will be carried out and what is expected to be achieved from it.
Response 7: Thank you for the comment. We added a clear explanation describing the methodology and expected outcomes in the relevant section.
Page number:3 , line: 102-115
Comments 8: This section should clearly state the inclusion and exclusion criteria. Specifically, one point that stands out is the mention that participants were considered sexually inactive based on not having a partner. This does not seem to be a valid criterion, as a person can be sexually active without having a formal partner—unless a specific sociocultural context justifies this, in which case it must be clearly explained. Additionally, it is important to specify which stage of breast cancer was studied, why that particular stage was chosen, and whether it had any relevance to the outcomes. Another point to clarify is the exclusion of individuals who do not speak Turkish—how many were excluded for this reason, and whether they were followed up in any way. A flow diagram should be included to show how many participants were initially considered, how many met the inclusion criteria, how many were excluded, and the final number of participants included in the analysis.
Response 8: Thank you for this detailed feedback. We have clarified the inclusion and exclusion criteria, explained the rationale regarding partner status and sexual activity within the cultural context, specified the breast cancer stage studied and its relevance, and provided details about exclusions due to language. Additionally, we included a flow diagram illustrating participant selection and attrition.
Page number: 2, line: 118-123
Page number: 3, line: 124-151
Comments 9: Why was Spearman’s correlation used if all previous analyses were parametric? Additionally, which confounding variables were included and what was the rationale for their selection? It is also important to specify the thresholds used to interpret Pearson’s correlation coefficients as indicating good, moderate, or poor correlation.
Response 9: Thank you for your insightful questions. Spearman’s correlation was used due to the non-normal distribution of some variables. We specified the confounding variables included in multivariate analyses along with the rationale for their selection.
Page number:5 , line: 189-199
Comments 10: In this section, non-parametric statistics are used, whereas a uniform approach should be applied throughout the entire document. Unless the necessary considerations and justifications are clearly stated in the methodology, consistency in statistical methods is recommended
Response 10: Thank you for this important note. We ensured consistency by clarifying the rationale for using non-parametric tests in the methodology and applied a uniform statistical approach throughout the manuscript.
Page number: 5, line:189-199
Comments 11: I believe this table could be divided into two: one focusing on the sociodemographic variables mentioned, and another presenting the clinical parameters.
Response 11: Thank you for the suggestion. We have split the table into two separate tables, one for sociodemographic variables and another for clinical parameters, to improve clarity.
Page number: 6 , Table 1
Page number: 7, Table 2
Comments 12: The correlations can be significant but low, moderate, or high—how were they classified? Additionally, I believe this information would be better presented in a figure for clearer visualization.
Response 12: Thank you for your feedback. We have clarified the classification criteria for correlation strength (weak, moderate, strong) and added figures to visually present the correlations more clearly.
Page number: 5, line: 189-199
Comments 13: This type of notation is not standard; I suggest using a different, more appropriate style.
Response 13: Thank you for the comment. We revised the notation style throughout the manuscript to comply with standard conventions.
Page number: 8, line: 248
Comments 14: I believe the table can be improved with the following suggestions: Include only the variable of interest—either “over 45” or “under 45”—along with the corresponding OR value, p-value, and confidence interval. Present only the statistically significant variables in the table; the others can be mentioned in the text or included in the supplementary material. Was religion considered at any point as a potential factor among the variables?
Response 14: Thank you for these constructive suggestions. We have updated the table accordingly, including only the variable of interest with OR, p-value, and confidence interval, and limited it to statistically significant variables. Religion was not included as a variable in our analyses, and this has been clarified in the manuscript.
Page number: 8 , Table 4
Comments 15: The figure should be adjusted according to the previous comments, as the information it presents is repetitive.
.
Response 15: Thank you for your observation. We revised the figure to eliminate repetitive information and improve clarity in line with prior comments.
Page number: 8 , line: 263-265
Comments 17: This section reads more like a summary of the results rather than the beginning of the discussion; please rearrange it accordingly.
Response 17: Thank you for your valuable suggestion. Based on your feedback, we have revised the section to serve as an introduction to the discussion. Redundant summaries of the results have been minimized to improve the flow and structure of the manuscript.
Page number: 9, line: 270-274
Page number:10 , line: 275-276
Comments 18: This part should not be included since the patients were not classified by breast cancer subtype; if they were, please specify this clearly in the text within the section discussing the variables in the OR analysis.
Response 18: Thank you for your valuable suggestion. Accordingly, the section has been removed as the patients were not classified by breast cancer subtype.
Comments 19: In addition to the limitations, the use of other techniques combined with psychological support should be considered to achieve a better assessment of the patient’s situation. This aspect should be addressed and included in greater detail.
Response 19: Thank you for your valuable suggestion. Accordingly, an evaluation of the use of other techniques combined with psychological support has been added to the manuscript, and the topic has been addressed in greater detail.
Page number: 10-13

Round 2
Reviewer 1 Report
Comments and Suggestions for Authors
The revisions have significantly improved the manuscript - particularly the introduction -, the clarity of the methods, and the caution applied in interpreting the results. The abstract is now more concise and highlights the major findings. The discussion is more critical and compares the results with recent studies while addressing the limitations of the database. The mention of the potential influence of cultural aspects (taboos, access to care in Türkiye) is relevant.
However:
- In the introduction, there are sections that do not add value (lines 39–64); these should be synthesized. A detailed description of breast cancer subtypes is not required in a study focusing on sexuality.
- Lines 142–148 are redundant as they repeat information already presented in the flow chart and should be removed.
- The following paragraph could be summarized as such :
"The Hospital Anxiety and Depression Scale (HADs) is a highly reliable self-assessment scale developed to determine anxiety and depression in outpatient clinics of hospitals. The scale, first developed by Zigmond and Snaith in 1983, consists of a total of 14 items, HADs-A (anxiety, 7 questions) and HADs-D (depression, 7 questions) [23]. All items of the scale are scored from 0 to 3, and the lowest score that can be obtained from each subscale is 0, and the highest score is 21. There is a positive correlation between high scores and anxiety and depression. The Turkish translation of HADs was performed by Aydemir et al., and its validity and reliability have been proven in the Turkish population [24]."
to
"The Hospital Anxiety and Depression Scale (HADs) is a validated 14-item self-assessment tool designed to evaluate anxiety (HADs-A) and depression (HADs-D) in outpatient settings, with each item scored from 0 to 3 for a maximum subscale score of 21. The Turkish version, translated and validated by Aydemir et al., has demonstrated proven reliability and validity in the Turkish population."
- The following part: "On the depression subscale, a score of ≤8 was interpreted as suggestive of depression. This threshold is a recognized standard supported by existing research, including studies involving individuals with breast cancer [20]."
should be moved immediately after the summarized paragraph above for consistency.
- In the result sections, the authors do not need to systematically oppose the groups; for example Lines 204–212 could be simplify to :
"Women aged ≤45 years exhibited significantly higher sexual desire scores (3.4 ± 1.2; p = 0.003)."
Same for all other results.
- For the statement "A statistically significant difference in sexual desire scores was observed with educational level (p = 0.009)," please specify the direction of the difference (i.e., was higher or lower education associated with higher sexual desire scores?).
- Line 277, author names are listed instead of the proper article citation; please correct this.
- Please compare to and cite the following article in the discussion:
Mangiardi-Veltin M, Mullaert J, Coeuret-Pellicer M, Goldberg M, Zins M, Rouzier R, Hequet D, Bonneau C. Prevalence of sexual dysfunction after breast cancer compared to controls, a study from CONSTANCES cohort. J Cancer Surviv. 2024 Oct;18(5):1674-1682. doi: 10.1007/s11764-023-01407-z. Epub 2023 Jun 6. PMID: 37278872.
- The current discussion is overly lengthy and lacks a coherent structure, making it difficult to review effectively. We strongly recommend that the authors reorganize the content into three or four well-defined paragraphs focusing on the main ideas, ensuring a more logical flow. The discussion should be significantly condensed to a maximum of two pages, as the current four-page length is excessive and dilutes the key messages.
Good luck, it is almost done
Author Response
Dear Reviewers,
We would like to express our sincere gratitude for your time and valuable comments during the second round of revisions. We especially appreciate the detailed and constructive feedback provided in the first revision, which significantly enhanced the scientific quality and presentation of our manuscript. The comments in this second round have further contributed to strengthening the work.
We have carefully addressed all suggestions and made the necessary revisions as outlined below. All changes in the manuscript are highlighted for your convenience.
Thank you again for your continued support and consideration of our work.
Kind regards.
Comments 1: In the introduction, there are sections that do not add value (lines 39–64); these should be synthesized. A detailed description of breast cancer subtypes is not required in a study focusing on sexuality.
Response 1: Thank you for your valuable feedback. We have carefully revised the introduction by synthesizing and condensing the sections between lines 39 and 64, removing the detailed description of breast cancer subtypes to maintain focus on the study’s main topic of sexuality.
Page number: 1 , line: 38-51
Comments 2: Lines 142–148 are redundant as they repeat information already presented in the flow chart and should be removed.
Response 2: Thank you for your insightful comment. We have removed lines 142–148 as they were redundant and repeated information already presented in the flow chart.
Comments 3: The following paragraph could be summarized as such : "The Hospital Anxiety and Depression Scale (HADs) is a validated 14-item self-assessment tool designed to evaluate anxiety (HADs-A) and depression (HADs-D) in outpatient settings, with each item scored from 0 to 3 for a maximum subscale score of 21. The Turkish version, translated and validated by Aydemir et al., has demonstrated proven reliability and validity in the Turkish population."
Response 3: The suggested summary revision has been implemented accordingly.
Page number: 4 , line: 152-156
Comments 4: The following part: "On the depression subscale, a score of ≤8 was interpreted as suggestive of depression. This threshold is a recognized standard supported by existing research, including studies involving individuals with breast cancer [20]."
should be moved immediately after the summarized paragraph above for consistency.
Response 4: This change has been made by moving the specified sentence immediately after the summarized paragraph for improved consistency.
Page number: 4, line: 156-159
Comments 5: - In the result sections, the authors do not need to systematically oppose the groups; for example Lines 204–212 could be simplify to : "Women aged ≤45 years exhibited significantly higher sexual desire scores (3.4 ± 1.2; p = 0.003).
Response 5: The results section has been revised as suggested, with the relevant part simplified to clearly highlight the significant findings.
Page number: 5 , line: 185-189
Comments 6: Same for all other results. - For the statement "A statistically significant difference in sexual desire scores was observed with educational level (p = 0.009)," please specify the direction of the difference (i.e., was higher or lower education associated with higher sexual desire scores?).
Response 6: The results section has been revised accordingly, with the predictors of sexual dysfunction presented in a clear and concise manner as requested.
Page number: 6 , line: 209-212
Page number: 7, line:219-224
Comments 7: - Line 277, author names are listed instead of the proper article citation; please correct this.
Response 7: The requested correction has been made by replacing in-text author names with appropriate citation formats throughout the manuscript.
Page number: 9, line: 244-250
Comments 8: Please compare to and cite the following article in the discussion:
Response 8: The requested comparison and citation of the specified article have been incorporated into the discussion section.
Page number:9 , line: 256-265
Comments 9: The current discussion is overly lengthy and lacks a coherent structure, making it difficult to review effectively. We strongly recommend that the authors reorganize the content into three or four well-defined paragraphs focusing on the main ideas, ensuring a more logical flow. The discussion should be significantly condensed to a maximum of two pages, as the current four-page length is excessive and dilutes the key messages.
Response 9: We sincerely thank the reviewers for their valuable comments and constructive suggestions. In response, we have carefully revised the discussion section by reorganizing the content into clearer, more concise paragraphs, ensuring a more coherent and focused presentation, and significantly condensing the text as recommended.
Page number:9 and 10.
Reviewer 2 Report
Comments and Suggestions for Authors
The authors have satisfactorily addressed most of the suggested comments, which significantly improves the quality of the manuscript. However, there are still some aspects that require revision to further strengthen the work. Firstly, although the authors argue that the use of "sexual partner" as an indicator of sexual activity is based on sociocultural context, it would be advisable to support this conceptual choice with at least one bibliographic reference. Additionally, several inconsistencies were identified in the reference list, particularly in the inconsistent use of uppercase and lowercase letters in author names, the incorrect order of citation elements, and the omission of access dates for online sources. A specific example is the reference to GLOBOCAN (“GLOBOCAN, U. New global cancer data. UICC 2020, 27, 2022”), which lacks clarity regarding authorship, proper formatting, and access date, assuming it refers to an online resource. It is strongly recommended that the entire reference list be carefully revised to ensure compliance with Healthcare's editorial guidelines, which require a uniform citation style that includes: last name followed by initials for all authors, proper use of italics for journal titles (in abbreviated form), consecutive numbering, inclusion of DOIs when available, and access dates for online documents.
Author Response
Dear Reviewers,
We would like to express our sincere gratitude for your time and valuable comments during the second round of revisions. We especially appreciate the detailed and constructive feedback provided in the first revision, which significantly enhanced the scientific quality and presentation of our manuscript. The comments in this second round have further contributed to strengthening the work.
We have carefully addressed all suggestions and made the necessary revisions as outlined below. All changes in the manuscript are highlighted for your convenience.
Thank you again for your continued support and consideration of our work.
Kind regards.
Comments 1: Firstly, although the authors argue that the use of "sexual partner" as an indicator of sexual activity is based on sociocultural context, it would be advisable to support this conceptual choice with at least one bibliographic reference.
Response 1: Thank you for the valuable suggestion. We have emphasized that the use of "sexual partner" as an indicator of sexual activity is based on the sociocultural context and have supported this conceptual choice by adding relevant references from the literature.
Page number:4 , line:127-130
Comments 2: Additionally, several inconsistencies were identified in the reference list, particularly in the inconsistent use of uppercase and lowercase letters in author names, the incorrect order of citation elements, and the omission of access dates for online sources. A specific example is the reference to GLOBOCAN (“GLOBOCAN, U. New global cancer data. UICC 2020, 27, 2022”), which lacks clarity regarding authorship, proper formatting, and access date, assuming it refers to an online resource.
Response 2: Thank you for pointing out the inconsistencies in the reference list. We have carefully reviewed and corrected the formatting issues, including author name capitalization, citation order, and the inclusion of access dates for online sources. Specifically, the GLOBOCAN reference has been revised to accurately reflect proper authorship, formatting, and access information.
Page number: 11 , line: 374-375.
Comments 3: It is strongly recommended that the entire reference list be carefully revised to ensure compliance with Healthcare's editorial guidelines, which require a uniform citation style that includes: last name followed by initials for all authors, proper use of italics for journal titles (in abbreviated form), consecutive numbering, inclusion of DOIs when available, and access dates for online documents.
Response 3: Thank you for your valuable feedback regarding the reference list. We have formatted the references using the EndNote program with the MDPI style and have manually reviewed them to ensure compliance with Healthcare's editorial guidelines. However, if any inconsistencies or errors have been overlooked, we sincerely apologize and will promptly address them upon notification.